# From Tradition to Health: Chemical and Bioactive Characterization of Five Traditional Plants

**DOI:** 10.3390/molecules27196495

**Published:** 2022-10-01

**Authors:** Paula Garcia-Oliveira, Anxo Carreira-Casais, Eliana Pereira, Maria Inês Dias, Carla Pereira, Ricardo C. Calhelha, Dejan Stojković, Marina Sokovic, Jesus Simal-Gandara, Miguel A. Prieto, Cristina Caleja, Lillian Barros

**Affiliations:** 1Centro de Investigação de Montanha (CIMO), Instituto Politécnico de Bragança, Campus de Santa Apolónia, 5300-253 Bragança, Portugal; 2Laboratório Associado para a Sustentabilidade e Tecnologia em Regiões de Montanha (SusTEC), Instituto Politécnico de Bragança, Campus de Santa Apolónia, 5300-253 Bragança, Portugal; 3Nutrition and Bromatology Group, Department of Analytical and Food Chemistry, Faculty of Food Science and Technology, University of Vigo-Ourense Campus, E-32004 Ourense, Spain; 4Institute for Biological Research "Siniša Stanković"-National Institute of Republic of Serbia, University of Belgrade, Bulevar despota Stefana 142, 11000 Belgrade, Serbia

**Keywords:** medicinal plants, phenolic composition, bioactive compounds, health benefits

## Abstract

Several scientific studies have been proving the bioactive effects of many aromatic and medicinal plants associated with the presence of a high number of bioactive compounds, namely phenolic compounds. The antioxidant, anti-inflammatory, and antimicrobial capacities of these molecules have aroused high interest in some industrial sectors, including food, pharmaceuticals, and cosmetics. This work aimed to determine the phenolic profiles of the infusions and hydroethanolic extracts of five plants (*Carpobrotus edulis*, *Genista tridentata*, *Verbascum sinuatum*, *Cytisus multiflorus*, and *Calluna vulgaris*) that have been employed in many traditional preparations. In addition, the antioxidant, antimicrobial, anti-inflammatory, and anti-tumoral activity of each different preparation was evaluated using in vitro assays. The HPLC-DAD-ESI/MS profile revealed the presence of eighty phenolic compounds, belonging to seven different families of compounds. Regarding antioxidant properties, the hydroethanolic extract of *C. edulis* showed a potent effect in the TBARS assay (IC_50_ = 1.20 µg/mL), while *G. tridentata* hydroethanolic extract achieved better results in the OxHLIA test (IC_50_ = 76 µg/mL). For cytotoxic and anti-inflammatory results, *V. sinuatum* infusions stood out significantly, with GI_50_ = 59.1–92.1 µg/mL and IC_50_ = 121.1 µg/mL, respectively. Finally, *C. edulis* hydroethanolic extract displayed the most relevant antibacterial activity, showing MBC values of 0.25–1 mg/mL, while *G. tridentata* hydroethanolic extract exerted the greatest antifungal effects (MFC of 0.5–1 mg/mL). The results of this study deepen the knowledge of the phenolic profiles and also provide evidence on the bioactive properties of the species selected, which could be considered highly valuable options for research and application in several sectors, namely food, cosmetics, and pharmaceuticals.

## 1. Introduction

In traditional or folk medicine, numerous plant species have been employed in different preparations, mainly in the form of infusions [1]. The beneficial properties on consumers’ health have been linked to the presence of diverse bioactive compounds, which have caught the attention of the pharmaceutical industry. Several traditional plants have been used as a natural source of phytochemicals by the pharmaceutical industry [1,2,3], for example, aspirin (obtained from *Salix* sp.), opium-derived morphine (extracted from *Papaver somniferum*), or the anticancer compound paclitaxel (derived from *Taxus brevifolia*) [4]. On the other hand, the nutraceutical and cosmetics industries are also interested in bioactive compounds to enhance the properties of their formulations. Therefore, the bioactive compound characterization of numerous traditional plants has been carried out. Among bioactive compounds, polyphenols are some of the most studied. To date, tens of thousands of different polyphenols have been identified, being the largest family of compounds found in plants and derived products. These compounds have been described to exert many beneficial properties, such as antioxidant, anti-inflammatory, anticancer, antimicrobial, or anti-diabetic properties, among others, as demonstrated by in vitro, in vivo, and clinical trials in humans [5,6,7]. In recent decades, these diverse biological properties, together with the absence of toxicity, have prompted the use of phenolic compounds as natural ingredients and additives in food, cosmetics, and pharmaceutical products, increasing their added value and improving biological features [8].

In this study, we studied several plant species that have been employed in traditional medicine. *Calluna vulgaris* L., (Scotch heather or ling), is a plant native to Europe and North Africa that has been employed in folk medicine to prepare infusions and decoctions to treat diseases such as depression, urinary tract infections, and inflammatory infections [9,10]. In addition, different extracts have shown antimicrobial, antioxidant, anti-inflammatory, anxiolytic, and anti-depressant effects, which are attributed to the presence of diverse bioactive compounds, especially phenolic compounds [11,12]. *Genista tridentata* L. (prickly broom), found in the Iberian Peninsula and northern Morocco, has been usually used in infusions or decoctions for the treatment of inflammatory and respiratory diseases, urinary disorders, and malaria and also as a diuretic, tonic, sedative, and cicatrizing agent [13,14]. Regarding biological properties, different extracts have also been shown to exert antioxidant and anti-inflammatory properties [14,15]. Some studies support their strong bioactivity with the presence of bioactive compounds discovered in their phytochemical composition, especially flavonoids [13,15]. *Cytisus multiflorus* (L’Hér.) Sweet (white broom) is an endemic plant of the Iberian Peninsula. According to ethnopharmacological studies, its fresh or dried flowers have been employed in infusions, decoctions, and tonics for the treatment of different disorders, such as inflammatory diseases, diabetes, migraine, or cutaneous eruptions, due to its anti-inflammatory, anti-diabetic, anti-hypertension, and diuretic properties [16,17]. The antioxidant properties of *C. multiflorus* extracts have been reported, but the other biological properties of this plant have not been deeply studied [16]. *Verbascum sinuatum* L. (mullein) can be found in southern Europe and northern Africa and Iran. Infusions of the flowers and leaves have been used to treat respiratory disorders and urinary infections and also as wound-healing agents [18,19]. Previous studies have reported the antioxidant and antimicrobial activity of this plant, which could be related to the presence of high contents of phenolic compounds, especially flavonoids [20,21]. Finally, *Carpobrotus edulis* L. (ice plant, hottentot-fig or sour fig), commonly found in coastal areas, is native to South Africa but has spread along the coasts of temperate areas around the world due to its invasive nature. In Africa, this species has been used in the folk treatment of human immunodeficiency virus infections, dysentery, skin wounds, throat infections, or digestive problems, among others [22,23]. Several studies have reported the antioxidant, antiproliferative, antimicrobial and antiviral activities of this plant [24,25,26].

The aim of this study was to evaluate the phenolic profile and the bioactive properties (namely, antioxidant, antimicrobial, anti-inflammatory and antitumor activity) of the infusions and hydroethanolic extracts obtained from five plant species employed in traditional medicine (*C. vulgaris*, *G. tridentata*, *C. multiflorus*, *V. sinuatum* and *C. edulis*). This study provides a deep characterization of the phenolic profile of the selected plants and also explores the biological properties of some species that have not been previously evaluated. Comparing both preparations allows us to obtain information on the possibilities of these plants as a source of compounds with bioactive potential that could be used in further bio-based applications.

## 2. Results and Discussion

### 2.1. Identification of the Main Phenolic Composition

The peak characteristics (retention time, wavelength of maximum absorption and mass spectral data), tentative identification and quantification (mg/g of extract) of the phenolic compounds present in the hydroethanolic extracts and infusion preparations of *C. vulgaris*, *G. tridentata*, *C. multiflorus*, *V. sinuatum*, and *C. edulis*, are presented in Table 1.

The illustrative phenolic profiles of the extracts, recorded at 280 and 370 nm, are shown in Figure 1. Overall, eighty phenolic compounds were tentatively identified in the five plants under study: 39 *O*-glycosylated flavonoids, 16 phenolic acids, 11 isoflavonoids, 6 *C*-glycosylated flavonoids, 4 flavan-3-ols, 2 iridoid glycosides and 2 phenylethanoid glycosides. In the literature, the phenolic profiles of *C. vulgaris* [10,12,27,28], *G. tridentata* [15,29], *C. multiflorus* [17,30], *V. sinuatum* [20] and *C. edulis* [26,31,32] have been descried. These studies were used for the identification and description of the phenolic compounds presented below, as well as others with different plant matrices (when not existing for the plant in question). For an easier description of the eighty phenolic compounds detected, the descriptions have been divided in the following section.

#### 2.1.1. *O*-Glycosylated Flavonoids

The biological importance of flavonoids is unquestionable. In addition to all the functions that they have in plants and other natural matrices (e.g., stress tolerance, protection against herbivores), they also exert health-promoting effects on the human organism [33]. Flavonoids are the most abundant group of phenolic compounds in plants, usually found linked to sugar moieties via *O*- or *C*- linkages [34]. The glycosylation is the most significant structural feature of flavonoids that determines their pharmacokinetic behavior and also the parent compound’s bioavailability since it modifies the overall structural polarity of the molecule [33].

Quercetin *O*-glycosylated derivatives had a high presence in the studied samples, with us having found fourteen compounds of this category. Peak 15^cv^ was identified as quercetin-3-*O*-glucoside by comparing the retention time, mass spectra data, and UV-vis spectra with the available standard compound. Peaks 10^g^, 11^g^, 14^g^, 16^cv^, 18^cv^, and 6^cm^ presented the deprotonated ion as peak 15^cv^ ([M-H]^−^ at *m*/*z* 463), λmax ranging from 353 to 354 nm, and one major MS^2^ peak at *m*/*z* 301 ([M-H-162]^−^) that corresponded to quercetin aglycone and the loss of a hexosyl unit (162 u), were all tentatively identified as quercetin-*O*-hexoside. Peak 1^cm^ presented a higher deprotonated ion [M-H]^−^ at *m*/*z* 625 and two MS^2^ peaks at *m*/*z* 463 and 301 ([M-H-162-162]^−^) that corresponded to the successive losses of two hexosyl units and were all tentatively identified as quercetin-*O*-dihexoside. Peak 19**^cv^** ([M-H]^−^ at *m*/*z* 447) presented the MS^2^ base peak at *m*/*z* 301, implying the loss of a deoxyhexosyl moiety (146 u), and was tentatively identified as quercetin-*O*-deoxyhexoside. Peaks 22^cv^ and 23^cv^, besides the loss of the deoxyhexosyl moiety (146 u), also presented the loss of 42 u (acyl group) from the deprotonated ion ([M-H]^−^ at *m*/*z* 449) until one of the major MS^2^ peaks at *m*/*z* 447 and were both tentatively identified as quercetin-acyl-*O*-deoxyhexoside. Finally, peaks 8^g^, 9^g^ and 5^cm^ presented a deprotonated ion ([M-H]^−^) at *m*/*z* 609 and one major MS^2^ peak at *m*/*z* 301 ([M-H-146-162]^−^), the loss of a deoxyhexosyl followed by the hexosyl moiety, and were tentatively identified as quercetin-*O*-deoxyhexosyl-hexoside. The presence of lower amounts of quercetin linked to one hexosyl moiety has been previously described in *C. vulgaris* methanolic extracts and infusion preparations, reporting 0.35 and 0.77 mg/g extract. On the other hand, higher amounts were found in the purified methanol/formic acid fraction of *C. vulgaris,* reaching a maximum concentration of 18.7 mg/g extract [10]. The compound quercetin-deoxyhexoside has also been found, in very low amounts, in the hydroethanolic extracts of *C. vulgaris* flowers from Portugal (101.30 µg/g extract) [12]. Regarding the samples of *C. multiflorus*, *O*-glycosylated quercetin derivatives have also been previously identified in a previous study [30]. Hydroethanolic extracts of samples from Portugal showed lower content values for quercetin linked to two hexosyl moieties (0.43 ± 0.05 mg/g dry weight) and higher contents of quercetin linked to rhamnosyl and hexosyl moieties (4.06 ± 0.34 mg/g dry weight) than those described herein.

One isorhamnetin derivative was found in *C. vulgaris* samples, the peak 20^cv^ ([M-H]^−^ at *m*/*z* 505), which could not be completely identified. It was only tentatively identified as an isorhamnetin derivative. The organic and aqueous extracts of *C. vulgaris* [10], and also its purified acetone fractions [28], have already been described as a source of this type of compound linked to hexosyl, deoxyhexosyl, and even phenolic acid moieties.

Two *O*-glycosylated derivatives of isorhamnetin were also found in *C. multiflorus* samples, peaks 8^cm^ and 9^cm^, [M-H]^−^ at *m*/*z* 623 and 477, respectively, presenting both one major MS^2^ base peak at *m*/*z* 315 that corresponded to the loss of 146 u + 162 u (deoxyhexosyl-hexosyl moieties) and 162 u (hexosyl moiety), respectively, being tentatively identified as isorhamnetin-*O*-deoxyhexosyl-hexoside and isorhamnetin-*O*-hexoside, respectively.

The plant sample that revealed the highest number of *O*-glycosylated compounds was *C. edulis* (quantitatively, this was *G. tridentata*, with a total amount of 59 ± 1 mg/g extract), which is in agreement with previous studies [26,31,32]. *C. edulis* also presented isorhamnetin derivatives, peaks 12^ce^/16^ce^ ([M-H]^−^ at *m*/*z* 623) and 9^ce^ ([M-H]^−^ at *m*/*z* 769) with the MS^2^ major fragment at *m*/*z* 315, tentatively identified as isorhamnetin-*O*-deoxyhexosyl-hexoside and isorhamnetin-*O*-deoxyhexosyl-hexosyl-*O*-deoxyhexoside, respectively. As reported by a previous study, laricitrin derivatives are also one of the major flavonoids found in *C. edulis* [26]. Four derivatives were found in the present study, peaks 11^ce^/10^ce^ ([M-H]^−^ at *m*/*z* 639) and 8^ce^ ([M-H]^−^ at *m*/*z* 771), revealing MS^2^ fragments coherent with those previously stated by the same study [26]. Therefore, they were tentatively identified as laricitrin-*O*-deoxyhexosyl-hexoside and laricitrin-*O*-pentosyl-*O*-deoxyhexosyl-hexoside, respectively. Peak 9^ce^ ([M-H]^−^ at *m*/*z* 947) presented a similar MS^2^ fragmentation pattern as peak **8^ce^**, but the *m*/*z* 771 fragment and its molecular ion gave the indication of the linkage of a hexuronosyl unit (176 u) to the molecule, so it was tentatively identified as laricitrin-*O*-hexuronosyl-*O*-pentosyl-*O*-deoxyhexosyl-hexoside. As far as the authors’ knowledge, this is the first time that this compound has been reported in *C. edulis* samples.

Despite the presence of other flavonoids, syringetin derivatives were the most abundant flavonoids in *C. edulis*, not only qualitatively, but also quantitatively (representing about 60% of the total flavonoids found in both hydroethanolic extracts and infusion preparations). It has been proven that syringetin enhances radiosensitivity in cancer cells [35] and also stimulates osteoblast differentiation resulting in bone formation [36], and, therefore, *C. edulis* could be a potential focus for medical studies in this field. Eight syringetin derivatives were found in *C. edulis* samples. The tentative identification of peaks **17^ce^/18^ce^** ([M-H]^−^ at *m*/*z* 653), **15^ce^** ([M-H]^−^ at *m*/*z* 639), and 13^ce^/14^ce^ ([M-H]^−^ at *m*/*z* 785) as syringetin-*O*-deoxyhexosyl-hexoside, syringetin-*O*-pentosyl-hexoside and syringetin-*O*-pentosyl-*O*-deoxyhexosyl-hexoside, respectively, was performed accordingly to previous authors [26,32]. Peak 20^ce^ ([M-H]^−^ at *m*/*z* 549) presented a characteristic MS^2^ fragmentation with the fragment at *m*/*z* 345 as the major base beak (syringetin aglycone), corresponding to the loss of 204 u ([M-H-162-42]^−^), tentatively identified as syringetin-*O*-acetyl-hexoside. Finally, peaks 22^ce^/21^ce^ presented a deprotonated ion [M-H]^−^ at *m*/*z* 815, and the main MS^2^ fragments at *m*/*z* 653 ([M-H-162]^−^) and *m*/*z* 345 ([M-H-162-146]^−^), corresponding to the loss the consecutive loss of a hexosyl, a deoxyhexosyl and another hexosyl unit, and were tentatively identified as syringetin-*O*-hexosyl-*O*-deoxyhexosyl-hexoside.

*C. multiflorus* samples presented two apigenin *O*-glycosylated derivatives, peaks 11^cm^ ([M-H]^−^ at *m*/*z* 431) and 12^cm^ ([M-H]^−^ at *m*/*z* 473), with one major fragment ion MS^2^ at *m*/*z* 269 (apigenin aglycone), corresponding to the loss of hexosyl ([M-H-162]^−^) and acyl-hexosyl ([M-H-42-162]^−^) moieties, respectively, tentatively identified as apigenin-*O*-hexoside and apigenin-*O*-acylhexoside, respectively. Particularly, peak 11^cm^ has already been identified in ethanolic extracts of *C. multiflorus* from Castro D’Aire Portugal in very similar concentrations as those (0.8 ± 0.1 mg/g dried plant) obtained herein for the hydroethanolic extracts [17]. Moreover, two luteolin derivatives were found in *C. multiflorus* samples, peaks 11^cm^ and 12^cm^, both tentatively identified as luteolin-*O*-malonyl-hexoside. Both presented a deprotonated ion [M-H]^−^ at *m*/*z* 533 and two major MS^2^ fragments at *m*/*z* 489 (44 u) and 285 (42 u + 162 u); the *m*/*z* 285 corresponded to the luteolin aglycone (identified by the characteristic UV spectra of the I band of the flavonol structure), the consecutive loss of 44u + 42 u corresponded to the malonyl moiety, and, finally, 16u corresponded to the hexosyl moiety. Previous studies found luteolin derivatives in ethanolic extracts of *C. multiflorus* [17,30]. However, none of the derivatives were equal to the ones found in the present study. The geographical location, the edaphic–climatic conditions of production, the year of collection, the type of extraction and solvents used can all be factors for these differences.

The chrysoeriol derivatives were only found in *G. tridentata* samples, peaks 13^g^ and 15^g^, presenting both a deprotonated ion [M-H]^−^ at *m*/*z* 461 and one major MS^2^ base peak at *m*/*z* 299 (chrysoeriol aglycone, [M-H-162]^-^), tentatively identified as chrysoeriol-*O*-hexoside. The previous reports on *G. tridentata*’s phenolic profile [15,29,37] do not describe the existence of this molecule in the samples studied. The beneficial medicinal effects of this molecule have already been proven by several authors [38,39], so, despite not being one of the major compounds in the plant, the presence of chrysoeriol derivatives increases its academic interest.

Finally, one kaempferol derivative was found in *C. vulgaris* samples (peak 21^cv^, the kaempferol derivative was differentiated from luteolin by the characteristic UV spectra), presenting a deprotonated ion [M-H]^−^ at *m*/*z* 431 and one major MS^2^ peak at *m*/*z* 285 (kaempferol aglycone, loss of 146 u), tentatively identified as kaempferol-*O*-deoxyhexoside [10,28].

#### 2.1.2. Phenolic Acids

Phenolic acids represent the second major family of phenolic compounds found in the five samples studied. Mostly were hydroxycinnamic acid derivatives (caffeic, quinic, *p*-coumaric and ferulic derivatives). These compounds are one of the most important classes of phenolic compounds since they are involved in many bioactive properties. They have been widely known as potent antioxidants and they have been linked with the prevention of oxidative stress-related diseases, such as cancer and cardiovascular and neurological disorders. Some of them also possess anti-inflammatory, anticancer, antimicrobial, and anti-diabetic properties, among others [40].

Ellagic acid hexoside (peak 12^g^) was only found in *G. tridentata*, presenting a deprotonated ion at [M-H]^−^ at *m*/*z* 463 and an MS^2^ fragment at *m*/*z* 301 that corresponded to the loss of 162 u (hexosyl unit). Ellagic acid was identified with the characteristic λmax at 260/331 nm and also the characteristic MS^3^ fragment at *m*/*z* 256 and 185.

In *V. sinuatum* samples, two phenolic acids were identified, quinic acid (peak 1^vs^) and caffeic acid hexoside (2^vs^), using the data previously described in *Verbascum ovalifolium* Donn ex Sims [41]. Peak 5^cv^, found in *C. vulgaris* samples for the first time, was tentatively identified as *p*-coumaric hexoside based on the deprotonated ion presented [M-H]^-^ at *m*/*z* 325 and the MS^2^ fragment at *m*/*z* 163 ([coumaric acid-H]^−^, −162 u), corresponding to the coumaric acid structure and the loss of a hexosyl unit [42]. Moreover, for the first time in *C. edulis*, the compound feruloyl-hexoside (peak 4^ce^) was tentatively identified, presenting a deprotonated molecule [M-H]^−^ at *m*/*z* 355 and one major MS^2^ fragment at *m*/*z* 193 ([ferulic acid-H]^−^, −162 u, loss of a hexosyl moiety), corresponding to the ferulic acid structure and the loss of a hexosyl unit [43].

As previously stated, hydroxycinnamic acid derivatives represent the major group of phenolic acids found, despite being found in only two of the studied samples, *C. edulis* and *C. vulgaris*. Two caffeoylquinic acids were identified, 3-*O*- (peaks 1^cv^ and 2^cv^) and 5-*O*-caffeoylquinic acids (peaks 2^ce^ and 3^ce^), both presenting a deprotonated ion [M-H]^−^ at *m*/*z* 353 and *m*/*z* 191 as the most MS^2^ prominent ion, having been differentiated by the retention time and by the hierarchical keys proposed by some studies [44,45], in which the authors differentiate the two compounds by the intensity of the fragment at *m*/*z* 179 (relative and weak intensity for the 3-*O*- and 5-*O*-derivatives, respectively). Since the peaks 1^cv^/2^cv^ and peaks 2^ce^/3^ce^ presented the same chromatographic characteristics, they were assigned as *cis* and *trans* isomers, respectively.

The same hierarchical keys were used to identify the *p*-coumaroylquinic acid derivatives. Peaks 1^ce^, 4^cv^, 5^ce^, 6^ce^, 6^cv^ and 7^cv^ all presented the same deprotonated ion [M-H]^−^ at *m*/*z* 337 but the differentiation of the isomers was easier than for the caffeoylquinic acid derivatives since the MS^2^ base peak at *m*/*z* 163 allowed the identification of 3-*O*-*p*-coumaroylquinic acid (peaks 1^ce^/4^cv^); *m*/*z* 191 allowed the identification of 5-*O*-*p*-coumaroylquinic acid (peaks 5^ce^ and 6^ce^, *cis* and *trans* forms, respectively) and *m*/*z* 173 allowed the identification of 4-*O*-*p*-coumaroylquinic acid (peaks 6^cv^ and 7^cv^, *cis* and *trans* forms, respectively). Finally, peak 7^ce^ ([M-H]^−^ at *m*/*z* 367) was tentatively identified as 5-*O*-feruloylquinic acid, presenting an MS^2^ base peak at *m*/*z* 191, which is again in accordance with the hierarchical keys proposed by previous studies for the LC-MS^n^ differentiation of hydroxycinnamic acids [44,45].

There are not many studies that prove the presence of this group of compounds in *C. edulis* and *C. vulgaris*. A previous study reported the presence of one hydroxycinnamic acid derivative, 1,3-*O*-dicaffeoylquinic acid, in the hydromethanolic extracts of *C edulis*. from South Africa, but it did not show quantification results for this compound [31]. On the other hand, a study performed with *C. vulgaris* from Portugal revealed the presence of 5-*O*-caffeoylquinic acid in purified acetone extracts at a maximum concentration of 1.29 mg/g extract; values lower than those found in the present study, in which the sum of the entire set of hydroxycinnamic derivatives achieved the concentrations of 3.67 and 8.2 mg/g extract in the hydroethanolic extracts and infusion preparations, respectively [28].

#### 2.1.3. Isoflavonoids

Eleven isoflavonoids were identified in the samples: myricetin, genistein, eriodictyol, and naringenin derivatives, but also puerarin derivatives. This group of compounds was only found in two samples studied: *G. tridentata* and *C. vulgaris*. In this latter species, these compounds represented the majority of the phenolic compounds (45% and 36% in the hydroethanolic extract and infusion preparations, respectively). Four myricetin derivatives were found—peaks 9^cv^ and 12^cv^ presented the same deprotonated ion [M-H]^−^ at *m*/*z* 479 and only one major MS^2^ fragment at *m*/*z* 317 (myricetin aglycone) corresponded to the loss of a hexosyl moiety ([M-H-hexosyl]^−^, 162 u), and so were tentatively identified as myricetin-*O*-hexoside. Similar behavior was observed for peak 14^cv^; however, the deprotonated ion [M-H]^−^ at *m*/*z* 463 and the MS^2^ fragment at *m*/*z* 317 revealed only the loss of 146 u (deoxyhexosyl moiety), allowing the tentative identification of myricetin-*O*-deoxyhexoside. These compounds have been previously reported in acetone extracts of *C. vulgaris* [28] and in hydroethanolic extracts of *C. vulgaris* flowers [12]. In the particular case of the study performed by Mandim et al., in purified acetone extracts of *C. vulgaris*, the contents obtained for myricetin-*O*-deoxyhexoside are lower (4.9 mg/g extract) than those obtained in the present study (6.7 and 5.97mg/g extract in the hydroethanolic extract and infusion, respectively); the same was not observed for myricetin-*O*-hexoside, since the authors reported a maximum content of 9.8 mg/g extract [28].

The last myricetin derivative was found in the *G. tridentata* sample, tentatively identified as myricetin-*C*-hexoside (peak 3^g^), presenting a deprotonated ion [M-H]^−^ at *m*/*z* 479 and characteristic MS^2^ fragments at *m*/*z* 359, 341, 221 and 167, coherent in a *C*-glycosyl linkage between the sugar moiety and the isoflavonoid structure [15]. This compound was the major flavonoid found in the *G. tridentata* sample, representing one-quarter of the total isoflavonoids found (15.037 ± 0.498 and 9.807 ± 0.222 mg/g extract in the hydroethanolic extract and infusion, respectively). Myricetin has been widely recognized as a compound with numerous health-promoting bioactivities that prove the diverse therapeutic purposes of this compound, such as its potential against diabetes mellitus [46], cardiovascular diseases [47] and obesity [48].

The second major group of isoflavonoids found was genistein derivatives. This group was only observed in *G. tridentata* samples. Peak 4^g^ presented a deprotonated ion [M-H]^−^ at *m*/*z* 593 and two MS^2^ fragments at *m*/*z* 413 (loss of 162 u) and at *m*/*z* 269 (loss of an additional 162 u), coupled with the λmax at 259 nm (characteristic of isoflavonoids), allowing the tentative identification of the isoflavonol aglycone as genistein, and therefore the molecule as genistein-*O*-dihexoside. This compound was already identified in *G. tridentata* flowers, presenting relatively higher values (1.43 mg/g extract) than those presented herein for the hydroethanolic extracts (1.00 mg/g extract) and infusion preparations (0.77 mg/g extract) [29]. The authors also reported the presence of a genistein derivative ([M-H]^−^ at *m*/*z* 413, peaks 5^g^ and 6^g^). Therefore, this reference was used as the basis for the tentative identification of these peaks. As far as the authors’ knowledge, there are no references in the literature that would allow a more complete identification of these compounds.

Finally, peak 16^g^ showed a deprotonated ion [M-H]^−^ at *m*/*z* 473 and two main MS^2^ fragments at *m*/*z* 311 and at *m*/*z* 269. As there is no bibliographic reference in *G. tridentata* about this compound, the chromatographic data of *Pueraria lobata* (Willd.) Ohwi were used, from which the authors tentatively identified this compound as *O*-acetylgenistein [37].

Peak 7^g^, also found in *G. tridentata*, presented a deprotonated molecule at [M-H]^−^ at *m*/*z* 431 and a prominent MS^2^ fragment at *m*/*z* 311 that corresponded to the loss of 120 u and the characteristic mass loss of *C*-glycosylated molecules, and was tentatively identified as hydroxy-puerarin (*C*-glycosylated form of the isoflavone daidzin) by comparison with the mass fragmentation and UV spectra responses reported in the literature [49,50]. This is the first time that this compound is reported in *G. tridentata* samples.

The two remaining isoflavonoid compounds were only found in *C. vulgaris* samples. Peaks 13^cv^ and 17^cv^ showed a deprotonated molecule at [M-H]^−^ at *m*/*z* 449 and 433, respectively, and a unique MS^2^ fragment at *m*/*z* 287 (eriodictyol aglycone) and *m*/*z* 271 (naringenin aglycone), respectively, that corresponded to the loss of a hexosyl moiety in both cases, tentatively identifying them as eriodictyol-*O*-hexoside and naringenin-*O*-hexoside, respectively. The compound eriodictyol-*O*-hexoside has been reported in the purified acetone fractions of *C. vulgaris* in slightly higher amounts (1.10 mg/g extract) [28] than those reported herein for the hydroethanolic extracts and infusion preparations (0.33 and 0.9 mg/g extract, respectively).

#### 2.1.4. *C*-Glycosylated Flavonoids

The *C*-glycosylation of flavonoid compounds is restricted to certain types of plants, and therefore they become a group of compounds of high academic and industrial value. Chemically, the *C*-glycosidic linkage of the saccharide moiety and flavonoid carbon skeleton significantly alters the pharmacokinetics and bioactivities of these compounds, when compared to *O*-glycosylated flavonoids, since it protects the cleavage of the molecule from the hydrolytic effect caused by acidic and enzymatic treatments [34].

In this study, this group of compounds was only found in *C. multiflorus* samples, which is in agreement with previous reports [17,30], and also in *G. tridentata* samples [15,29]. The identification of the peaks 2^cm^, 3^cm^/4^cm^ and 7^cm^ was based on the mass spectra results reported by these two authors. Peak 2^cm^ ([M-H]^−^ at *m*/*z* 579) was tentatively identified as 2″-*O*-pentosyl-8-*C*-hexoside luteolin [17]; peaks 3^cm^/4^cm^ ([M-H]^−^ at *m*/*z* 563) were tentatively identified as 2″-*O*-pentosyl-8-*C*-hexoside apigenin isomer I and isomer II, respectively [17,30]; and, finally, peak 7^cm^ ([M-H]^−^ at *m*/*z* 707) was tentatively identified as 6’’-*O*-(3-hydroxy-3-methylglutaroyl)-2’’-*O*-pentosyl-*C*-hexosyl-apigenin [17].

Several authors have stated that the *C*-glycosylated flavonoids were always in higher amounts than the *O*-glycosylated flavonoids in plants [34]. This statement was confirmed in the present study, as the *C*-glycosyl derivatives represent 72.2% and 76.5% of the total amount of phenolic compounds in the hydroethanolic extracts (115 mg/g extract) and infusions (116 mg/g extract) of *C. multiflorus*, respectively. The high amount of these compounds is mainly due to the presence of 2’’*-O*-pentosyl-8-*C*-hexoside luteolin, peak 2^cm^, with a content of 62 mg/g of extract and 59 mg/g of extract in the hydroethanolic extracts and infusion preparations, respectively. It is also worth noting that the amount of this compound is much higher than the amount obtained in previous studies [17,30]. The contents of 2″-*O*-pentosyl-8-*C*-hexoside, peaks 3^cm^/4^cm^, are in accordance with the literature [30].

Finally, two *C*-glycosylated quercetin derivatives were found in *G. tridentata* samples, peaks 1^g^/2^g^, presenting both a deprotonated ion at [M-H]^−^ at *m*/*z* 465 and characteristic MS^2^ fragments of *C*-glycosyl at *m*/*z* 447, 375, 357, 345 (major base peak), 327, 317 and 167, consistent with that previously reported by Caleja et al. (2019), tentatively identified as dihydroquercetin 6-*C*-hexoside isomer I and isomer II, respectively. Comparing the findings with previous studies, higher amounts of this compound have been reported in infusions of *G. tridentata* (45 ± 1 mg/g extract) [29].

#### 2.1.5. Flavan-3-Ols

This family was exclusively found in *C. vulgaris* samples, having been previously described in acetone extracts [28] and in the hydroethanolic extracts of *C. vulgaris* flowers [12]. Peak 3^cv^ presented a deprotonated ion [M-H]^−^ at *m*/*z* 305 and characteristic MS^2^ fragments at *m*/*z* 219, *m*/*z* 179 and *m*/*z* 125, coherent with an (epi)gallocatechin molecule, as previously described [51]. For the identification of peaks 10^cv^ and 8^cv^/11^cv^, the information obtained by mass spectrometry allowed the identification and the relative position of the elementary units of the molecules, but since it is not possible to observe the main fragmentation mechanisms of proanthocyanidins (retro-Diels-Alder (RDA), quinone methide (QM), heterocyclic ring fission (HFR), and benzofuran-forming (BFF) mechanisms), it was not possible to identify the position between the (epi)catechin units and the differentiation of catechin isomers. As such, 10^cv^ ([M-H]^−^ at *m*/*z* 591) and 8^cv^/11^cv^ ([M-H]^−^ at *m*/*z* 863) were tentatively identified as A-Type proanthocyanidins and *β*-type (Epi)catechin trimers, respectively, comparing the mass spectral data with a previous report [52]. Some authors have reported, in hydroethanolic extracts of *C. vulgaris* flowers, the content of *β*-type (Epi)catechin trimer to be 11.6 times lower than the highest reported herein (2.9 ± 0.2 mg/g of extract in the infusion of *C. vulgaris*) [12]. However, a different scenario was observed in purified acetone fractions of *C. vulgaris*, in which the authors obtained a maximum concentration of 45 mg/g of fraction, indicating a concentration 15.5 higher than the herein obtained for *β*-type (Epi)catechin trimer compound [28].

#### 2.1.6. Phenylethanoid Glycosides

This family was exclusively found in *V. sinuatum* samples. Two phenylethanoid glycosides were found, peaks 3^vs^ and 4^vs^, presenting a deprotonated ion [M-H]^−^ at *m*/*z* 623 and MS^2^ fragments at *m*/*z* 461, *m*/*z* 315, *m*/*z* 179, *m*/*z* 161 and *m*/*z* 153, corresponding to the loss of a caffeoyl, caffeoyl-rhamnosyl, caffeate, anydro-caffeate and hydroxytyrosol, respectively, as described previously in *Verbascum ovalifolium* [41], in addition to the characteristic hydroxycinnamic acid derivatives UV spectra at 325–327 nm, being for that manner tentatively identified as verbascoside and isoverbascoside, respectively. Both peaks have been already found in *V. sinuatum* samples from Iran, but no quantitative data were provided [20]. In the studied samples of *V. sinuatum*, verbascoside was the major compound found, representing 36.4% and 49.2% of the total phenolic compounds found in the hydroethanolic extract (13.1 ± 0.3 mg/g extract) and infusion (12.4 ± 0.4 mg/g extract), respectively.

#### 2.1.7. Iridoid Glycosides

As in the previous case, this family was exclusively found in *V. sinuatum* samples. Two iridoid glycosides, tentatively identified as *p*-Coumaroyl-6-*O*-rhamnosyl aucubin isomer I and isomer II (peaks 5^vs^ and 6^vs^, respectively), were described according to the phenolic data of *V. ovalifolium* [41]. Both peaks presented a deprotonated ion [M-H]^−^ at *m*/*z* 637 and MS^2^ fragments at *m*/*z* 475, *m*/*z* 309 and *m*/*z* 205 that correspond to the previously described.

Peaks 5^vs^ represent the second major compound found in *V. sinuatum* samples, with contents of 6.4 ± 0.1 and 2.7 ± 0.1 mg/g extract in the hydroethanolic extract and infusion, respectively. To our knowledge, this is the first description of these compounds in *V. sinuatum* samples.

### 2.2. Biological Properties

#### 2.2.1. Antioxidant Studies

The results of the antioxidant activity of the hydroethanolic extracts and infusions of the selected plants are shown in Table 2. For the determination of the antioxidant activity, two assays were performed, the OxHLIA assay and the TBARS assay, which evaluate oxidative hemolysis and the inhibition of lipid peroxidation, respectively. Due to the multifactorial behavior of the antioxidant activity, it is essential to determine such bioactivity through different methodologies. In the case of the TBARS assay, the results showed that hydroethanolic extracts presented a higher activity than infusions, with values ranging from 1.2 to 10.2 and 5.3–51 µg/mL for each preparation, respectively. Specifically, regarding the extracts of the plants, all displayed a lower IC_50_ value than Trolox (23 ± 2 µg/mL), showing a potent antioxidant activity. *C. edulis* and *G. tridentata* hydroethanolic extracts displayed the best results, with 1.20 ± 0.05 and 3.19 ± 0.02 µg/mL, respectively. These results could be attributed to the presence of phenolic acids in the samples (7.1 ± 0.1 and 10.2 ± 0.4 mg/g extract), which are well known for their in vitro and in vivo antioxidant activity through radical scavenging mechanisms [53]. Compared with previous studies, lower TBARS IC_50_ values were obtained in the present study. For example, a study using *G. tridentata* flowers collected in Trás-os-Montes, Portugal, reported an IC_50_ of 8.2 mg/mL of infusion, where no phenolic acids were detected [29]. Other authors reported an IC_50_ of 92 mg/mL for an infusion of *Pterospartum tridentatum* (synonym of G. tridentata) flowers [54]. These variations could be attributed to environmental conditions (temperature, soil, light, etc.), which may significantly affect the chemical composition and biological properties of plants [55]. Regarding *C. edulis*, to our knowledge, no study has analyzed its antioxidant properties by TBARS assay, but several studies have evaluated them by 1,1-diphenyl-2-picrylhydrazyl (DPPH), nitric oxide and scavenging assays, ferric reducing power, chelating ability, inhibition of linoleic acid peroxidation, and *β*-carotene bleaching assays [23,24]. For the OxHLIA analysis, few positive results were obtained. Neither *V. sinuatum* nor *C. vulgaris* preparations showed activity. The infusion of *C. multiflorus* achieved a value of 109 ± 9 µg/mL, while the value for the hydroethanolic extract of *C. edulis* was 132 ± 6 µg/mL. The infusion and hydroethanolic extract of *G. tridentata* achieved similar activity, with 78 ± 6 and 76 ± 5 µg/mL, respectively, showing higher activity than that of Trolox (85 ± 2 µg/mL). In the literature, values lower than those of the present study have been reported (IC_50_ = 37.7 g/mL of infusion) [29], but no studies have focused on hydroethanolic extracts. In general, these results suggest that *C. edulis* and *G. tridentata* could be sources of natural antioxidants.

#### 2.2.2. Cytotoxic and Anti-Inflammatory Studies

The results of cytotoxicity, hepatotoxicity, and anti-inflammatory activities are shown in Table 2. According to the cytotoxic evaluation, determined by sulforhodamine B assay, all plant preparations showed inhibitory effects against the selected cancer cell lines, but the results were not comparable with those of the positive control, ellipticine (GI_50_ values of 0.91–3.2 µg/mL). Specifically, *V. sinuatum* showed the highest activity, with GI_50_ values varying between 59.1 and 92.1 mg/mL of infusion and 101.1–172.2 µg/mL of hydroethanolic extract, HeLa being the most sensitive cell line in both preparations. *G. tridentata* preparations also showed significant cytotoxic activity, with GI_50_ values of 83.2–142.7 and 102.9–160.5 μg/mL for the infusion and hydroethanolic extract, respectively. As observed in *V. sinuatum*, HeLa cells were the most susceptible to both preparations. Hydroethanolic extracts of *C. vulgaris* showed potent cytotoxic activity against HeLa and HepG2 cell lines, with GI_50_ values of 69.6 and 79.4 µg/mL, respectively, but much higher GI_50_ values were observed in the other cancer cell lines. No hepatotoxicity was observed in the samples, except for *V. sinuatum* infusion (GI_50_ values of 223.1 ± 15.4 μg/mL). Despite this result, it is observed that it is more effective against cancer cells than against non-cancer cells. Regarding anti-inflammatory activity, the IC_50_ values oscillated between 121.1 and 293.2 μg/mL and 130.1–237.9 μg/mL for infusions and hydroethanolic extracts, respectively. However, the results were not comparable with dexamethasone (IC_50_ 16 ± 1 μg/mL). No anti-inflammatory activity was observed in hydroethanolic extracts of *C. vulgaris* and *C. multiflorus* and the infusion of *C. edulis*. As in the previous analysis, *V. sinuatum* preparations showed better results, with IC_50_ values varying between 121.1 ± 3.9 and 130.1 ± 2.8 μg/mL for the infusion and hydroethanolic extract, respectively.

To our knowledge, no scientific studies have evaluated the cytotoxic and anti-inflammatory properties of *V. sinuatum* preparations. These properties could be attributed to the high content of phenylethanoid glycosides (14.9 ± 0.4 mg/g and 15.4 ± 0.3 for the infusion and hydroethanolic extract, respectively). These compounds have been recently described to exert many different biological properties, including anticancer and anti-inflammatory activities [56,57]. Although the extract contains more phenylethanoid glycosides than the infusion, this is the one that presents the best results. This may be due to the presence of other polar compounds present in the sample and synergy phenomena between compounds. Considering these results, *V. sinuatum* could be a matrix with promising biological properties for further research.

#### 2.2.3. Antibacterial and Antifungal Studies

Table 3 displays the values for antibacterial and antifungal evaluation. Regarding antibacterial analysis, all samples exerted inhibitory and bactericidal activity. For infusions, MIC and MBC ranged between 0.25 and 4 mg/mL, while for hydroethanolic extracts, these parameters ranged from 0.25 to 2 mg/mL. Except in the case of *V. sinuatum*, hydroethanolic extracts exerted better results than infusions. The most susceptible strain was *S. aureus*, with MIC and MBC values around 0.25–05 mg/mL for both types of preparations. Nevertheless, none of the preparations displayed similar bactericidal effects to those reported for the positive control streptomycin, whose MBC values ranged between 0.05 and 0.3 mg/mL. Considering the results, the species with the most remarkable antibacterial activity was *C. edulis*, especially the hydroethanolic extract. In the first species, this preparation showed MIC and MBC values of 0.25 mg/mL against *S. aureus*, while values of 0.5 mg/mL were observed for *B. cereus*, *L. monocytogenes*, and *S.* Typhimurium. For *M. flavus*, MIC and MBC values were 0.5 and 1 mg/mL, respectively. Finally, the less susceptible stain was *E. cloacae*, with MIC and MBC values of 1 mg/mL. Previous studies have reported that different extracts of *C. edulis* leaves displayed antimicrobial effects against *B. cereus, S. aureus, Staphylococcus epidermidis*, and *Mycobacterium tuberculosis,* while little effect was observed against Gram-negative bacteria such as *E. coli, P. aeruginosa*, or *S.* Typhimurium [23,58]. In a recent study, the hydroethanolic (1:1, *v*/*v*) extract of *C. edulis* samples from Portugal did not show antimicrobial activity against *Klebsiella pneumonia, E coli*, or *S. aureus* [26], which does not agree with the results obtained (Table 3). According to some authors, different phenolic acids and flavonoids are involved in the antibacterial effects of this plant, together with other compounds such as terpenoids [23].

Similarly, for antifungal activity, all samples exerted quite potent inhibitory and fungicidal activity. For infusions, MIC and MFC ranged between 0.25 and 1 mg/mL and 0.5–1 mg/mL, respectively, while for hydroethanolic extracts, these parameters oscillated between 0.12 and 1 mg/mL and 0.25–2 mg/mL, respectively. In this case, infusions showed to be more effective for *C. edulis, C. multiflorus*, and *C. vulgaris*, while hydroethanolic extracts achieved better results for *G. tridentata* and *V. sinuatum*. The most susceptible fungi were *A. versicolor* and *A. fumigatus*. Specifically, the hydroethanolic extract of *G. tridentata* showed the lowest MIC and MFC. For *A. fumigatus*, *A. versicolor*, *A. niger*, and *P. funiculosum*, these values were 0.25 and 0.50 mg/mL, respectively. For *P. aurantiogriseum*, values of 0.5 and 1 mg/mL were reported, respectively. This preparation displayed comparable antifungal effects to those observed for ketoconazole, which showed MFC values of 0.50 mg/mL. These findings may suggest that *G. tridentata* could be explored as a source of antifungal compounds in future research. Among the literature, few studies have evaluated the antifungal properties of *G. tridentata*. Hydromethanolic extracts of this plant have been reported to inhibit several *Candida* species, showing inhibition halos between 9 and 14 mm [59]. Other authors described that *G. tridentata* infusion showed MIC and MFC values of 0.5 and 1 mg/mL, respectively, against *A. versicolor*, *P. funiculosus*, and *Penicillium verrucosum*. For *A. niger*, MIC values reached 8 mg/mL, while MFC was higher than 8 mg/mL [29]. Comparing the results of the present study, lower MIC and MFC were observed for hydroethanolic extracts, suggesting that fewer polar compounds are involved in this bioactivity, among which phenolic compounds could be involved. In this sense, flavonoids such as quercetin, myricetin, and genistein and their *O*-glycoside derivatives have been reported to exert antifungal properties [60]. As could be observed in Table 3, samples of *G. tridentata* contain significant amounts of these flavonoids and derivatives.

## 3. Materials and Methods

### 3.1. Samples

All plant materials were collected in the spring of 2019 (depending on each species’ phenology). *Genista tridentata*, *Cytisus multiflorus*, and *Calluna vulgaris* flowers were collected from the Natural Park of Montesinho territory, Trás-os-Montes, northeastern Portugal (41°53′29″ N 6°50′36″ O). *Verbascum sinuatum* leaves were collected in Villanueva del Río y Minas Seville province, southern Spain, (37°39’ N 5°42’ O). These samples were air-dried and stored in the freezer at −20 °C. On the other hand, *Carpobrotus edulis* was collected from Toralla Island, Vigo, Pontevedra Province, northwestern Spain 42°12′02″ N 8°48′00″ O. This species was first frozen at −80 °C and then lyophilized in a LyoAlfa 15, Telstar.

### 3.2. Preparations

The infusions and hydroethanolic extracts were prepared following protocols previously described [61]. For each infusion, 1 g of the dry sample was added to 200 mL of boiled, distilled water and left to stand at room temperature for 5 min. All the samples were filtered through Whatman no. 4 paper, frozen at −20 °C, and then lyophilized (FreeZone 4.5, Labconco, Kansas City, MO, USA) to obtain a dry extract. On the other hand, for preparing hydroethanolic extracts, 1 g of dried sample was added to 50 mL of ethanol/water (80:20 *v*/*v*). The mixture was left under stirring at room temperature for 1 h and then filtered. The residue was re-extracted with an additional 50 mL of the same solution, under the same conditions. Both extracts were evaporated at 40 °C in a rotary evaporator (100 rpm, 40 °C; rotary evaporator, Heidolph, Schwabach, Germany) to remove the alcoholic fraction. The aqueous phase was frozen and lyophilized to obtain a dry extract.

### 3.3. Phenolic Compound Composition

For the chromatographic analysis of the phenolic compounds, the lyophilized hydroethanolic extracts and infusion preparations were re-dissolved in an ethanol:water solution (20:80, *v*/*v*) in order to obtain a final solution of 10 mg/mL. The chromatographic analysis was performed according to that previously described by a previous work [62], using a Dionex Ultimate 3000 UPLC system (Thermo Scientific, San Jose, CA, USA) equipped with a quaternary pump, an automatic injector (at 5 °C), a degasser, and a column compartment with an automated thermostat. The detection of compounds was carried out with a diode detector (DAD), using wavelengths of 280 nm, 330 nm and 370 nm. For the separation of the compounds, a Waters Spherisorb S3 ODS-2 C_18_ reverse-phase column (4.6 × 150 mm, 3 μm; Milford, MA, USA) was used, thermostatted at 35 °C. The HPLC system described was also connected to a mass spectrometer (MS). The detection of MS was done using an Ion Trap Linear LTQ XL mass spectrometer (ThermoFinnigan, San Jose, CA, USA) equipped with an ESI source (electrospray ionization source). Data were collected and analyzed using the Xcalibur^®^ program (Thermo Finnigan, San Jose, CA, USA). For the identification of the compounds, the data obtained (retention times, UV-Vis spectra and mass spectra) were compared with data available in the literature and, when available, with the standards. For quantitative analysis, calibration curves were obtained by the injection of standard solutions with known concentrations (described in the corresponding tables), based on UV-Vis signals, and using the length of the maximum absorption wave of each standard compound. In the cases where there was no availability of standards for the respective compounds, the quantification was done through the calibration curves of compounds of the same phenolic group. The results were expressed in mg of compound per mg/g of extract.

### 3.4. Bioactive Evaluation

#### 3.4.1. Evaluation of Antioxidant Activity

All the dried samples were re-dissolved in water or ethanol/water solution (80:20 *v*/*v*) (2.5 mg/mL). To measure antioxidant activity, two assays were performed: inhibition of lipid peroxidation and oxidative hemolysis inhibition assays. The first one was performed according to a previously described method [61]. Briefly, the inhibition of lipid peroxidation in porcine (*Sus scrofa*) brain homogenates was evaluated through the decrease in thiobarbituric acid reactive substances (TBARS). The results were expressed in IC_50_ values (inhibitory concentration that gives 50% of antioxidant activity) in µg/mL. The oxidative hemolysis inhibition assay (OxHLIA) was carried out using sheep erythrocytes, by the method previously described [63]. The results were expressed in IC_50_ values (concentration with the ability to produce a Δt hemolysis delay of 60 min) in µg/mL. In both cases, the synthetic antioxidant Trolox was used as a positive control.

#### 3.4.2. Evaluation of the Cytotoxic and Hepatotoxic Activity

Cytotoxicity was evaluated in different human tumor cell lines: MCF-7 (adenocarcinoma of the breast), NCI-H640 (non-small cell of lung cancer), HeLa (carcinoma of the cervix), and HepG2 (hepatocellular carcinoma). In addition, a non-tumor cell line PLP2 (obtained from pig liver, purchased from certified slaughterhouses) was used to evaluate the cytotoxicity of the samples against healthy cells. To that aim, the dry extracts were dissolved in water at final concentrations of 6.25–400 μg/mL, and a sulforhodamine B assay was employed to measure the inhibition of the cell growth, as described previously [61]. Ellipticin was used as a positive control. The results were expressed as growth inhibitory concentration 50 (GI_50_), which represents the extract concentration inhibiting 50% of net cell growth, in μg/mL.

#### 3.4.3. Evaluation of Anti-Inflammatory Activity

The anti-inflammatory activity of the samples was evaluated in murine macrophages RAW 264.7 cell line subjected to lipopolysaccharide (LPS)-induced inflammation, which causes the production of nitric oxide (NO) by the cells. The inhibition of NO was determined using the Griess Reagent System Kit (Promega), as described previously [61]. Samples were resuspended in water at final concentrations of 6.25–400 μg/mL. The results were expressed as IC_50_, which represents the extract concentration, in μg/mL, that inhibits the production of NO by macrophages by 50%. Dexamethasone was used as a positive control.

#### 3.4.4. Evaluation of Antimicrobial Activity

The dried plant extracts were dissolved in water (10 mg/mL) and the antibacterial capacity was evaluated by the microdilution method, using the methodology previously described [64]. For the antibacterial assay, different Gram-positive bacteria were employed-*Staphylococcus aureus* (ATCC 6538), *Bacillus cereus* (clinical isolate), *Micrococcus flavus* (ATCC 10240), and *Listeria monocytogenes* (NCTC 7973)-as well as Gram-negative bacteria-*Enterobacter cloacae* (ATCC 35030) and *Salmonella typhimurium* (ATCC 13311). Minimum inhibitory concentration (MIC) and the minimum bactericidal (MBC) concentration, in mg/mL, were determined. Streptomycin was used as a positive control.

Antifungal activity was determined according to a previous protocol [65]. Five strains of fungi were employed: *Aspergillus fumigatus* (human isolate), *Aspergillus versicolor* (ATCC 11730), *Aspergillus niger* (ATCC 6275), *Penicillium funiculosum* (ATCC 26839), and *Penicillium aurantiogriseum* (ATCC 58604). MIC and minimal fungicidal concentration (MFC), in mg/mL, were calculated. Ketoconazole was used as a positive control.

The microorganisms employed for the analysis were deposited at the Mycological Laboratory, Department of Plant Physiology, Institute for Biological Research “Sinisa Stanković”, University of Belgrade, Serbia.

### 3.5. Statistical Analysis

All the tests were carried out in triplicate and the results were expressed as mean values and standard deviations (SDs). Significant differences between samples were analyzed using Student’s t-test with α = 0.05, using IBM SPSS Statistics for Windows, Version 23.0. (IBM Corp., Armonk, NY, USA).

## 4. Conclusions

The present study evaluated the phenolic profile of five plant species employed in traditional remedies. Eighty compounds were tentatively identified, belonging to seven families of compounds: 39 *O*-glycosylated flavonoids, 16 phenolic acids, 11 isoflavonoids, 6 *C*-glycosylated flavonoids, 4 flavan-3-ols, 2 iridoid glycosides and 2 phenylethanoid glycosides. *C. vulgaris* showed a significant content of iso- and *O*-glycosylated flavonoids. Similarly, in *G. tridentata*, the major compounds were *O*-glycosylated flavonoids. *C. multiflorus* showed the most prominent total flavonoid content, being especially rich in *C*-glycosylated flavonoids. In *V. sinuatum*, total phenylethanoid and iridoid glycosides represented more than 50% of the phenolic compounds identified. Finally, *O*-glycosylated flavonoids were the major compounds in *C. edulis*. Regarding bioactive properties, *C. edulis* and *G. tridentata* hydroethanolic extracts showed significant antioxidant properties. On the other hand, the infusion of V. sinuatum achieved promising results in cytotoxic and anti-inflammatory assays. Finally, *C. edulis* and *G. tridentata* hydroethanolic extract revealed considerable antibacterial and antifungal properties, respectively. Considering our findings, a deep characterization of phenolic compounds has been performed along with a diverse biological evaluation. Although more research is needed to identify the compounds involved in the biological properties, the present study suggests that some of these plants are interesting candidates for future research in bio-based applications

## Figures and Tables

**Figure 1 molecules-27-06495-f001:**
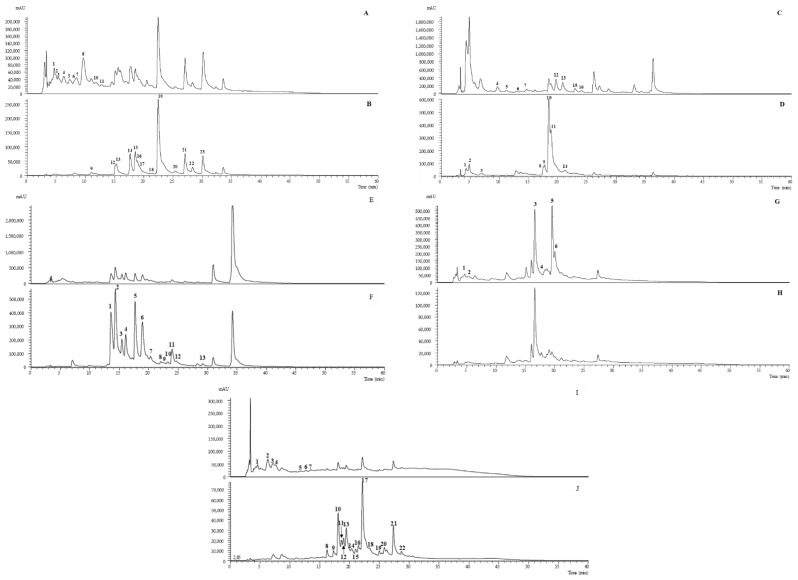
Illustrative phenolic profiles of *C. vulgaris* (**A**,**B**), *G. tridentata* (**C**,**D**), *C. multiflorus* (**E**,**F**), *V. sinuatum* (**G**,**H**), and *C. edulis* (**I**,**J**), recorded at 280 and 370 nm, respectively.

**Table 1 molecules-27-06495-t001:** Retention time (Rt), wavelengths of maximum absorption in the visible region (λmax), mass spectral data, identification, and quantification (mg/g extract) of the phenolic compounds present in the hydroethanolic extracts and infusions of *C. vulgaris*, *G. tridentata*, *C. multiflorus*, *V. sinuatum* and *C. edulis* (Mean ± SD), and the respective extraction yield (η, %).

Peak	Rt (min)	λmax (nm)	[M-H] ^−^(*m*/*z*)	MS^2^ (*m*/*z*)	Tentative Identification	Infusion	EtOH:W
*Calluna vulgaris*							
1 ^cv^	4.64	316	353	191(100),179(45),173(3),135(5)	*cis* 3-*O*-Caffeoylquinic acid	1.01 ± 0.06 *	0.65 ± 0.01 *
2 ^cv^	4.91	319	353	191(100),179(38),173(5),135(5)	*trans* 3-*O*-Caffeoylquinic acid	2.04 ± 0.1 *	0.46 ± 0.01 *
3 ^cv^	5.34	282	305	219(68),179(100),125(25)	(Epi)gallocatechin	2.5 ± 0.1 *	0.516 ± 0.02 *
4 ^cv^	6.29	297	337	191(7),173(5),163(100),155(5)	3-*O*-*p*-Coumaroylquinic acid	0.63 ± 0.03 *	0.27 ± 0.01 *
5 ^cv^	7.26	314	325	163(100),145(71),119(8)	*p*-Coumaric acid hexoside	0.62 ± 0.03 *	0.17 ± 0.01 *
6 ^cv^	8.47	305	337	191(7),173(100),163(15),155(5)	*cis* 4-*O*-*p*-Coumaroylquinic acid	1.08 ± 0.03 *	0.29 ± 0.01 *
7 ^cv^	9.52	311	337	191(6),173(100),163(12),155(5)	*trans* 4-*O*-*p*-Coumaroylquinic acid	2.783 ± 0.001 *	1.828 ± 0.004 *
8 ^cv^	10.97	279	863	739(92),713(59),695(100),577(69),575(49),425(14),407(10),289(6),287(12)	β-type (Epi)catechin trimer	2.9 ± 0.2 *	1.23 ± 0.03 *
9 ^cv^	11.77	279	479	317(100)	Myricetin-*O*-hexoside	5.3 ± 0.1 *	5.36 ± 0.03 *
10 ^cv^	12.82	279	591	573(19),465(62),451(5),439(100),421(33),289(12)	A-Type proanthocyanidin	2.72 ± 0.01 *	0.56 ± 0.04 *
11 ^cv^	14.4	281	863	739(92),713(59),695(100),577(69),575(49),425(14),407(10),289(6),287(12)	β-type (Epi)catechin trimer	3.2 ± 0.2 *	0.89 ± 0.01 *
12 ^cv^	15.16	270/310	479	317(100)	Myricetin-*O*-hexoside	5.9 ± 0.1 *	5.8 ± 0.1 *
13 ^cv^	15.93	287/310	449	287(100)	Eriodictyol-*O*-hexoside	0.9 ± 0.1 *	0.33 ± 0.03 *
14 ^cv^	17.63	348	463	317(100)	Myricetin-*O*-deoxyhexoside	5.97 ± 0.01 *	6.7 ± 0.2 *
15 ^cv^	18.51	353	463	301(100)	Quercetin-3-*O*-glucoside	1.34 ± 0.02 *	1.81 ± 0.02 *
16 ^cv^	18.87	353	463	301(100)	Quercetin-*O*-hexoside	1.09 ± 0.03 *	1.43 ± 0.05 *
17 ^cv^	20.48	282	433	271(100)	Naringenin-*O*-hexoside	0.316 ± 0.002	tr
18 ^cv^	21.33	353	463	301(100)	Quercetin-*O*-hexoside	0.937 ± 0.003 *	1.13 ± 0.01 *
19 ^cv^	22.41	349	447	301(100)	Quercetin-*O*-deoxyhexoside	4.104 ± 0.006 *	4.9 ± 0.1 *
20 ^cv^	25.33	348	505	445(34),315(100)	Isorhamnetin derivative	1.03 ± 0.01 *	1.13 ± 0.01 *
21 ^cv^	27.02	343	431	285(100)	Kaempferol-*O*-deoxyhexoside	1.64 ± 0.03 *	1.94 ± 0.02 *
22 ^cv^	28.31	347	489	447(31),301(100)	Quercetin-acyl-*O*-deoxyhexoside	1.25 ± 0.03 *	1.27 ± 0.02 *
23 ^cv^	30.04	348	489	447(31),301(100)	Quercetin-acyl-*O*-deoxyhexoside	1.4 ± 0.1 *	1.79 ± 0.05 *
					**Total Phenolic Acids (TPA)**	8.2 ± 0.2 *	3.67 ± 0.05 *
					**Total Flavan-3-ols (TF3O)**	11.3 ± 0.5 *	3.20 ± 0.05 *
					**Total Isoflavonoids (TiF)**	18.4 ± 0.2	18.2 ± 0.3
					**Total *O*-glycosylated Flavonoids (TOF)**	12.79 ± 0.03 *	15.3 ± 0.3 *
					**Total Phenolic Compounds (TPC)**	51 ± 1 *	40.4 ± 0.6 *
					**Extraction Yield (η, %)**	9.5	27.72
*Genista tridentata*							
1 ^g^	4.29	291/344sh	465	447(12),375(19),357(5),345(100),327(6),317(5),167(8)	Dihydroquercetin 6-*C*-hexoside isomer I	8.5 ± 0.3 *	10.5 ± 0.1 *
2 ^g^	4.81	292/345sh	465	447(11),375(15),357(10),345(100),327(8),317(5),167(8)	Dihydroquercetin 6-*C*-hexoside isomer II	12 ± 1 *	14.5 ± 0.2 *
3 ^g^	6.76	286	479	359(100),341(9),221(10),167(9)	Myricetin-*C*-hexoside	9.8 ± 0.2 *	15.04 ± 0.49 *
4 ^g^	9.66	259	593	431(100),269(13)	Genistein-*O*-dihexoside	0.77 ± 0.04 *	1.004 ± 0.056 *
5 ^g^	12.85	363	413	311(100),269(5)	Genistein derivative	tr	tr
6 ^g^	13.57	333	413	311(100),269(5)	Genistein derivative	0.049 ± 0.002 *	0.081 ± 0.001 *
7 ^g^	14.61	261/286	431	311(100),283(32)	Hydroxy-puerarin	1.2 ± 0.1 *	1.9 ± 0.1 *
8 ^g^	17.48	347	609	301(100)	Quercetin-*O*-deoxyhexosyl-hexoside	1.12 ± 0.01 *	1.39 ± 0.02 *
9 ^g^	17.64	350	609	301(100)	Quercetin-*O*-deoxyhexosyl-hexoside	1.24 ± 0.05 *	1.56 ± 0.04 *
10 ^g^	18.48	354	463	301(100)	Quercetin-*O*-hexoside	2.5 ± 0.1 *	7.3 ± 0.1 *
11 ^g^	18.84	354	463	301(100)	Quercetin-*O*-hexoside	1.9 ± 0.1 *	5.03 ± 0.07 *
12 ^g^	19.67	260/331	463	301(100), 256(5),185(12)	Ellagic acid hexoside	5.1 ± 0.4 *	10.2 ± 0.4 *
13 ^g^	20.79	261/298	461	299(100)	Chrysoeriol-*O*-hexoside	1.14 ± 0.03 *	1.3 ± 0.1 *
14 ^g^	21.22	335	463	301(100)	Quercetin-*O*-hexoside	1.08 ± 0.01 *	1.5 ± 0.1 *
15 ^g^	22.96	260/286	461	299(100)	Chrysoeriol-*O*-hexoside	0.97 ± 0.01 *	1.3 ± 0.1 *
16 ^g^	23.99	260/286	473	311(13),269(100)	*O*-acetylgenistein	tr	0.32 ± 0.02
					**Total Phenolic Acids (TPA)**	5.1 ± 0.4 *	10.2 ± 0.4 *
					**Total Isoflavonoids (TiF)**	2.1 ± 0.1 *	3.3 ± 0.1 *
					**Total *O*-glycosylated Flavonoids (TOF)**	41 ± 1 *	59 ± 1 *
					**Total Phenolic Compounds (TPC)**	48 ± 2 *	73 ± 1 *
					**Extraction Yield (η, %)**	9.09	21.90
*Cytisus multiflorus*							
1 ^cm^	13.6	354	625	463(100),301(50)	Quercetin-*O*-dihexoside	7.2 ± 0.3	7.01 ± 0.17
2 ^cm^	14.35	348	579	459(23),429(73),357(71),327(100),309(66)	2’’-*O*-Pentosyl-8-*C*-hexoside luteolin	59 ± 1 *	62 ± 1 *
3 ^cm^	15.48	340	563	443(5),413(100),323(6),311(7),293(86)	2″-*O*-Pentosyl-8-*C*-hexoside apigenin isomer I	8.7 ± 0.2 *	7.9 ± 0.2 *
4 ^cm^	16.07	336	563	443(4),413(100),323(5),311(11),293(79)	2″-*O*-Pentosyl-8-*C*-hexoside apigenin isomer II	9.8 ± 0.4 *	9.3 ± 0.3 *
5 ^cm^	17.69	353	609	301(100)	Quercetin-*O*-deoxyhexosyl-hexoside	1.9 ± 0.1 *	7.1 ± 0.4 *
6 ^cm^	18.94	249	463	301(100)	Quercetin-*O*-hexoside	6.0 ± 0.1	6.04 ± 0.03
7 ^cm^	20.32	340	707	563(34),413(43),293(10)	6’’-*O*-(3-hydroxy-3-methylglutaroyl)-2’’-*O*-pentosyl-C-hexosyl-apigenin	10.6 ± 0.4 *	3.6 ± 0.2 *
8 ^cm^	22.18	342	623	315(100)	Isorhamnetin-*O*-deoxyhexosyl-hexoside	1.9 ± 0.1 *	1.5 ± 0.1 *
9 ^cm^	23.33	335	477	315(100)	Isorhamnetin-*O*-hexoside	2.1 ± 0.1 *	1.35 ± 0.02 *
10 ^cm^	23.13	334	431	269(00)	Apigenin-*O*-hexoside	2.5 ± 0.1 *	2.7 ± 0.1 *
11 ^cm^	23.98	346	533	489(100),285(26)	Luteolin-*O*-malonyl-hexoside	1.39 ± 0.05 *	2.7 ± 0.1 *
12 ^cm^	24.7	347	533	489(100),285(26)	Luteolin-*O*-malonyl-hexoside	2.6 ± 0.1 *	1.461 ± 0.1 *
13 ^cm^	28.33	333	473	269(100)	Apigenin-*O*-acylhexoside	2.05 ± 0.02 *	1.9 ± 0.1 *
					**Total *O*-glycosylated Flavonoids (TOF)**	28 ± 1 *	32 ± 1 *
					**Total *C*-glycosylated Flavonoids (TCF)**	88.7 ± 0.4 *	83 ± 2 *
					**Total Flavonoid Compounds (TFC)**	116 ± 1	115 ± 3
					**Extraction Yield (η, %)**	11.86	31.73
*Verbascum sinuatum*							
1 ^vs^	4.63	275	191	173(100),111(15)	Quinic acid	1.02 ± 0.01 *	1.25 ± 0.03 *
2 ^vs^	5.21	328	341	179(100),161(14)	Caffeic acid hexoside	0.112 ± 0.003 *	0.281 ± 0.004 *
3 ^vs^	16.58	325	623	461(100),315(34),179(8),161(9),153(10)	Verbascoside	12.4 ± 0.4 *	13.1 ± 0.3 *
4 ^vs^	17.7	327	623	461(100),315(34),179(11),161(8),153(13)	Isoverbascoside	2.5 ± 0.1 *	2.41 ± 0.03 *
5 ^vs^	19.57	314	637	475(100)309(71),205(8)	*p*-Coumaroyl-6-*O*-rhamnosyl aucubin isomer I	2.7 ± 0.1 *	6.4 ± 0.1 *
6 ^vs^	20.01	312	637	475(100)309(65),205(10)	*p*-Coumaroyl-6-*O*-rhamnosyl aucubin Isomer II	1.29 ± 0.03 *	2.3 ± 0.1 *
					**Total Phenolic Acids (TPA)**	1.14 ± 0.01 *	1.53 ± 0.03 *
					**Total Iridoid Glycosides (TiG)**	4.01 ± 0.11 *	8.7 ± 0.2 *
					**Total Phenylethanoid Glycosides (TpG)**	14.9 ± 0.4 *	15.4 ± 0.3 *
					**Total Phenolic Compounds (TPC)**	25.2 ± 0.2 *	36 ± 1 *
					**Extraction Yield (η, %)**	20.29	34.34
*Carpobrotus edulis*							
1 ^ce^	6.27	308	337	191(10),163(100),119(13)	3-*O*-*p*-Coumaroylquinic acid	0.31 ± 0.01 *	1.49 ± 0.01 *
2 ^ce^	7.16	324	353	191(100),179(14),135(10)	*cis* 5-*O*-Caffeoylquinic acid	0.57 ± 0.02 *	1.91 ± 0.01 *
3 ^ce^	7.59	324	353	191(100),179(14),135(10)	*trans* 5-*O*-Caffeoylquinic acid	0.49 ± 0.02 *	1.8 ± 0.1 *
4 ^ce^	8.61	326	355	193(100),175(34),149(5)	Feruloyl-hexoside	0.108 ± 0.003 *	0.38 ± 0.01 *
5 ^ce^	11.78	308	337	191(100),163(10),119(13)	*cis* 5-*O*-*p*-Coumaroylquinic acid	0.11 ± 0.01 *	0.57 ± 0.03 *
6 ^ce^	12.66	310	337	191(100),163(10),119(13)	*trans* 5-*O*-*p*-Coumaroylquinic acid	0.09 ± 0.01 *	0.56 ± 0.01 *
7 ^ce^	13.48	314	367	193(5),191(100),173(8)	5-*O*-Feruloylquinic acid	0.097 ± 0.0001 *	0.357 ± 0.003 *
8 ^ce^	16.22	325	771	639(92),330(100),315(32),287(8)	Laricitrin-*O*-pentosyl-*O*-deoxyhexosyl-hexoside	0.87 ± 0.01 *	1.01 ± 0.01 *
9 ^ce^	17.3	322	769	623(23),315(100)	Isorhamnetin-*O*-deoxyhexosyl-hexosyl-*O*-deoxyhexoside	0.863 ± 0.004 *	0.99 ± 0.01 *
10 ^ce^	18.06	349	639	331(100),315(23),287(10)	Laricitrin-*O*-deoxyhexosyl-hexoside	0.95 ± 0.01 *	1.37 ± 0.02 *
11 ^ce^	18.63	349	639	331(100),315(23),287(10)	Laricitrin-*O*-deoxyhexosyl-hesxoside	0.88 ± 0.01 *	1.109 ± 0.002 *
12^ce^	19.02	357	623	315(100)	Isorhamnetin-*O*-deoxyhexosyl-hexoside	0.881 ± 0.003 *	1.12 ± 0.03 *
13 ^ce^	19.43	271/353	785	653(34),345(100),330(35),287(21)	Syringetin-*O*-pentosyl-*O*-deoxyhexosylhexoside	0.95 ± 0.01 *	1.22 ± 0.01 *
14 ^ce^	20.26	275/348	785	653(34),345(100),330(35),287(21)	Syringetin-*O*-pentosyl-*O*-deoxyhexosylhexoside	0.867 ± 0.002 *	1.08 ± 0.02 *
15 ^ce^	20.99	276/347	639	345(100),330(34),287(3)	Syringetin-*O*-pentosyl-hexoside	0.865 ± 0.002 *	1.04 ± 0.01 *
16 ^ce^	21.51	272/344	623	315(100)	Isorhamnetin-*O*-deoxyhexosyl-hexoside	0.855 ± 0.004 *	1.053 ± 0.004 *
17 ^ce^	22.14	275/357	653	345(100),330(35),287(21)	Syringetin-*O*-deoxyhexosyl-hexoside	0.992 ± 0.004 *	1.81 ± 0.01 *
18 ^ce^	23.41	276/350	653	345(100),330(35),287(21)	Syringetin-*O*-deoxyhexosyl-hexoside	0.859 ± 0.002 *	1.03 ± 0.01 *
19 ^ce^	25.02	279/339	947	771(34),639(92),331(100),315(32),287(8)	Laricitrin-O-hexuronosyl-O-pentosyl-O-deoxyhexosyl-hexoside	0.854 ± 0.001 *	1.021 ± 0.002 *
20 ^ce^	25.82	272/362	549	345(100),330(28),287(5)	Syringetin-*O*-acetyl-hexoside	0.855 ± 0.001 *	1.01 ± 0.02 *
21 ^ce^	27.33	275/341	961	799(34),345(100),330(23),287(10)	Syringetin-*O*-hexosyl-*O*-dideoxyhexosyl-hexoside	0.8811 ± 0.0004 *	1.28 ± 0.01 *
22 ^ce^	28.69	279/348	815	653(23),345(100),330(30),287(10)	Syringetin-*O*-hexosyl-*O*-deoxyhexosyl-hexoside	0.846 ± 0.003 *	1.011 ± 0.002 *
					**Total Phenolic Acids (TPA)**	1.77 ± 0.04 *	7.1 ± 0.1 *
					**Total *O*-glycosylated Flavonoids (TOF)**	13.27 ± 0.04 *	17.2 ± 0.1 *
					**Total Phenolic Compounds (TPC)**	15.04 ± 0.08 *	24.3 ± 0.3 *
					**Extraction Yield (η, %)**	36.67	16.67

Results are presented as mean ± standard deviation and the statistical differences between infusions and hydroethanolic extracts for each species were analyzed by Student’s *t*-test with significant differences of *p* < 0.005. Significant differences between preparations are signaled with *. Standard calibration curves used for quantification: *Calluna vulgaris (cv)*: (-)-catechin (*y* = 84950*x* − 23200, LOD = 0.17 μg/mL; LOQ = 0.68 μg/mL, peaks 3, 8, 10 and 11), chlorogenic acid (*y* = 168823*x* − 161172, LOD = 0.20 µg/mL; LOQ = 0.68 µg/mL, peaks 1 and 2), myricetin (*y* = 23287*x* − 581708, LOD = 61.21 µg/mL and LOQ = 185.49 µg/mL, peak 9), naringenin (*y* = 18433*x* + 78903, LOD = 0.17 µg/mL and LOQ = 0.81 µg/mL, peaks 13 and 17), *p*-coumaric acid (*y* = 301950*x* + 6966.7, LOD = 0.68 μg/mL and LOQ = 1.61 μg/mL, peaks 4, 5, 6 and 7), and quercetin-3-*O*-glucoside (*y* = 34843*x* − 160173, LOD = 0.21 µg/mL; LOQ = 0.71 µg/mL, peaks 15, 16, 18, 19, 20, 21, 22, and 23). *Genista tridentata* (*g*): Daidzin (*y* = 27652*x* + 29187, R² = 0.9996, LOD = 20.58 µg/mL; LOQ = 62.35 µg/mL, peak 7), ellagic acid ((*y* = 26719*x* − 317255, *R²* = 0.9986, LOD = 41.20 µg/mL and LOQ =124.84 µg/mL, peak 12), genistein (*y* = 64642*x* + 187360, LOD = 14.97 µg/mL and LOQ 45.37 g/mL, peaks 4, 5, 6, and 16), myricetin (*y* = 23287*x* − 581708, LOD = 61.21 µg/mL and LOQ = 185.49 µg/mL, peak 3), and quercetin-3-*O*-glucoside (*y* = 34843*x* − 160173, LOD = 0.21 µg/mL; LOQ = 0.71 µg/mL, peaks 1, 2, 8, 9, 10, 11, 13, 14 and 15). *Cytisus multiflorus* (*cm*) Apigenin-6-*C*-glucoside (*y* = 197337*x* + 30036, LOD = 0.19 µg/mL and LOQ = 0.63 µg/mL, peaks 3, 4 and 7), apigenin-7-*O*-glucoside (y = 10,683x − 45,794, LOD = 0.10 μg/mL and LOQ = 0.53 μg/mL, peaks 10 and 13), luteolin-6-*C*-glucoside (*y* = 4087.1*x* + 72589, LOD = 0.86 μg/mL; LOQ = 1.67 μg/mL, peak 2), and quercetin-3-*O*-glucoside (*y* = 34843*x* − 160173, LOD = 0.21 µg/mL and LOQ = 0.71 µg/mL, peaks 1, 5, 6, 8, 9, 11 and 12). *Verbascum sinuatum* (*vs*): Caffeic acid (*y* = 388345*x* + 406369, LOD = 0.78 μg/mL and LOQ = 1.97 μg/mL, peak 2), chlorogenic acid (*y* = 168823*x* − 161172, LOD = 0.20 µg/mL and LOQ = 0.68 µg/mL, peak 1), *p*-coumaric acid (*y* = 301950*x* + 6966.7, LOD = 0.68 μg/mL and LOQ = 1.61 μg/mL, peaks 5 and 6), and verbascoside (*y* = 124233*x* − 18873, LOD = 0.70 µg/mL and LOQ = 2.13 µg/mL, peaks 3 and 4). *Carpobrotus edulis (ce):* Chlorogenic acid (*y* = 168823*x* − 161172, LOD = 0.20 µg/mL; LOQ = 0.68 µg/mL, peaks 2 and 3), ferulic acid (*y* = 633126*x* − 185462, LOD = 0.20 μg/mL and LOQ = 1.01 μg/mL, peaks 4 and 7), *p*-coumaric acid (*y* = 301950*x* + 6966.7, LOD = 0.68 μg/mL and LOQ = 1.61 μg/mL, peaks 1, 5 and 6), and quercetin-3-*O*-glucoside (*y* = 34843*x* − 160173, LOD = 0.21 µg/mL; LOQ = 0.71 µg/mL, peaks 8 to 22).

**Table 2 molecules-27-06495-t002:** Antioxidant, cytotoxic and anti-inflammatory activities of infusions and hydroethanolic extracts of selected plants.

	*C. vulgaris*		*G. tridentata*		*C. multiflorus*		*V. sinuatum*		*C. edulis*		Positive control
	I	EtOH:W	I	EtOH:W	I	EtOH:W	I	EtOH:W	I	EtOH:W	
**Antioxidant** (**IC_50_, µg/mL)**											**Trolox**
**TBARS**	18 ± 1 *	10.2 ± 0.3 *	5.3 ± 0.1 *	3.19 ± 0.02 *	51 ± 3 *	3.7 ± 0.1 *	17.4 ± 0.9 *	4.2 ± 0.2 *	24.0 ± 0.8 *	1.20 ± 0.05 *	23 ± 2
**OxHLIA**	n.a	n.a	78 ± 6	76 ± 5	109 ± 9	n.a	n.a	n.a	n.a	132 ± 6	85 ± 2
**Cytotoxic and hepatotoxicity activity (GI_50_, µg/mL)**											**Ellipticine**
**NCI H460**	334.6 ± 9.1 *	219.7 ± 10.8 *	142.7 ± 5.3 *	160.5 ± 5.3 *	246.8 ± 5.6 *	314.3 ± 8.7 *	92.1 ± 3.9 *	140.0 ± 5.8 *	272.1 ± 24.9	256.4 ± 10.9	1.0 ± 0.1
**Hela**	270.3 ± 8.2 *	69.6 ± 6.6 *	83.2 ± 6.5	102.9 ± 10.6	133.3 ± 9.6	147.9 ± 5.1	59.1 ± 3.1 *	101.1 ± 4.9 *	341.7 ± 15.1 *	295.5 ± 16.2	1.91 ± 0.06
**MCF-7**	322.1 ± 4.3 *	205.3 ± 9.6 *	129.1 ± 6.3 *	146.8 ± 6.5 *	235.8 ± 8.5 *	278.7 ± 8.1 *	74.8 ± 3.7 *	125.9 ± 6.5 *	289.7 ± 12.3 *	260.5 ± 7.9 *	0.91 ± 0.04
**HepG2**	296.4 ± 22.2 *	79.4 ± 4.5 *	123.1 ± 19.1	132.4 ± 8.5	216.1 ± 11.8 *	263.4 ± 21.2 *	65.4 ± 2.9 *	172.2 ± 22.7 *	306.3 ± 27.1 *	210.1 ± 16.7 *	1.1 ± 0.2
**PLP2**	>400	>400	>400	>400	>400	>400	223.1 ± 15.4	>400	>400	>400	3.2 ± 0.7
**Anti-inflammatory activity (IC_50_, µg/mL)**											**DX**
	229.2 ± 7.9	>400	144.4 ± 2.2 *	207.4 ± 15.5 *	293.2 ± 11.8	>400	121.1 ± 3.9 *	130.1 ± 2.8 *	>400	237.9 ± 5.8	6 ± 1

Results are presented as mean ± standard deviation and the statistical differences between infusions and hydroethanolic extracts in each species were analyzed by Student’s t-test with significant differences of *p* < 0.005. Significant differences between preparations are signaled with *. Abbreviations: I, infusion; EtOH:W, hydroethanolic extract; n.a: no activity; DX, dexamethasone.

**Table 3 molecules-27-06495-t003:** Antimicrobial activity of infusions and hydroethanolic extracts of selected plants (mg/mL).

	*C. edulis*				*G. tridentata*				*V. sinuatum*				*C. multiflorus*				*C. vulgaris*				Controls			
	I		EtOH:W		I		EtOH:W		I		EtOH:W		I		EtOH:W		I		EtOH:W		St		Kt	
Bacteria	MIC	MBC	MIC	MBC	MIC	MBC	MIC	MBC	MIC	MBC	MIC	MBC	MIC	MBC	MIC	MBC	MIC	MBC	MIC	MBC	MIC	MBC	MIC	MBC
*Staphylococcus aureus*	0.25	0.50	0.25	0.25	0.25	0.25	0.25	0.50	0.25	0.50	0.25	0.25	0.25	0.50	0.25	0.25	0.25	0.50	0.50	0.50	0.1	0.2	-	-
*Bacillus cereus*	1.00	1.00	0.50	0.50	1.00	2.00	0.50	1.00	1.00	1.00	0.50	0.50	2.00	2.00	1.00	2.00	1.00	1.00	1.00	1.00	0.025	0.05	-	-
*Micrococcus flavus*	1.00	2.00	0.50	1.00	2.00	4.00	1.00	2.00	1.00	2.00	1.00	2.00	2.00	4.00	1.00	1.00	4.00	4.00	1.00	2.00	0.05	0.1	-	-
*Listeria monocytogenes*	0.50	1.00	0.50	0.50	1.00	2.00	0.50	0.50	1.00	1.00	1.00	2.00	0.50	1.00	0.50	1.00	2.00	4.00	1.00	1.00	0.2	0.3	-	-
*Enterobacter cloacae*	0.50	1.00	1.00	1.00	1.00	2.00	1.00	2.00	0.50	1.00	1.00	2.00	1.00	2.00	2.00	2.00	1.00	1.00	2.00	2.00	0.025	0.05	-	-
*Salmonella* Typhimurium	0.50	1.00	0.50	0.50	1.00	2.00	0.50	1.00	1.00	2.00	1.00	2.00	0.50	1.00	1.00	1.00	2.00	2.00	0.50	1.00	0.1	0.2	-	-
**Fungi**	MIC	MFC	MIC	MFC	MIC	MFC	MIC	MFC	MIC	MFC	MIC	MFC	MIC	MFC	MIC	MFC	MIC	MFC	MIC	MFC	MIC	MFC	MIC	MFC
*Aspergillus fumigatus*	0.50	1.00	0.50	1.00	1.00	1.00	0.25	0.50	0.50	1.00	0.25	0.50	0.50	0.50	0.25	0.25	1.00	1.00	0.50	0.50	-	-	0.2	0.5
*Aspergillus versicolor*	0.50	1.00	0.50	1.00	0.50	1.00	0.25	0.50	0.50	1.00	0.25	0.50	0.25	0.50	0.25	0.25	0.50	1.00	0.50	1.00	-	-	0.2	0.5
*Aspergillus niger*	0.50	1.00	0.50	1.00	0.50	1.00	0.25	0.50	0.50	1.00	0.50	1.00	0.25	0.50	0.25	0.25	0.50	1.00	1.00	2.00	-	-	0.2	0.5
*Penicillium funiculosum*	0.50	0.50	0.50	1.00	0.50	1.00	0.25	0.50	0.50	1.00	0.12	0.50	0.25	0.50	0.50	0.50	0.50	1.00	1.00	2.00	-	-	0.2	0.5
*Penicillium aurantiogriseum*	0.50	1.00	1.00	2.00	0.50	1.00	0.50	1.00	0.50	1.00	0.50	1.00	0.25	0.50	0.50	1.00	0.50	1.00	1.00	2.00	-	-	0.2	0.5

Abbreviations: I, infusion; EtOH:W, hydroethanolic extract; MIC, minimal inhibitory concentration; MBC, minimal bactericidal concentration; MFC, minimal fungicidal concentration; St, streptomycin; Kt, ketoconazole.

## Data Availability

Not applicable.

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
