# Peer review of "From Tradition to Health: Chemical and Bioactive Characterization of Five Traditional Plants"

_molecules, 2022, doi:10.3390/molecules27196495_

Round 1

Reviewer 1 Report

REVIEW COMMENTS

Title of the manuscript: From tradition to health: chemical and bioactive characterization of five traditional plants

Overall, the work is interesting, current and well written. However, it needs to be updated and improved in some information.

Abstract: Well written and suggests using more specific terms, for example, in line 39, the term “biological effects” is too broad and in line 40, “bio-based application in food”  Is it “functional foods”?

Keywords: Appropriate

Introduction: This is well written. However, it is better to refer to the latest reference instead of reference 1.

Material and methods: Well written and the Authors have used appropriate assays and procedures for the study.

Results and Discussion: Well written. The authors have collected more relevant data and they are presented in a well-comprehending manner.

Conclusion: Well written and it is going with research objectives.

Author Response

Thank you for your careful reading of the manuscript and for all your comments. All suggestions for changes were made to guarantee the improvement of the article.

Reviewer 2 Report

The scientific article " From tradition to health: chemical and bioactive characterization of five traditional plants" presents detailed results of research on five plants used in traditional medicine. The authors expanded the knowledge of these plants by conducting research on their composition, antioxidant properties or biocidal properties. The article is structured correctly, the subject matter is comprehensibly presented and the methodology is correctly described. The paper is suitable for publication in its present form. However, I have one minor question and remark.

1. Question about the concentration of ethanol for making extracts. It is commonly believed that, ethanol concentration of 60%v/v upwards is treated as a biocide. It would be worthwhile in the future to also check lower concentrations of ethanol and their effect on biocidal properties, as it may turn out that they too will have similar properties. A lower amount of ethanol helps reduce the cost of producing such a formulation and is much more friendly to our skin.

Good Job :)

Author Response

 Thank you for your careful reading of the manuscript and for all your comments. Effectively, reconciling the extraction of bioactive compounds using technologies and solvents considered to be cleaner and more eco-friendly is a concern that our working group always considers. In this way, we always try to use water, ethanol, or a mixture of both, as they are considered a safe and permissible choice even when their incorporation into products in the food and/or pharmaceutical sector is envisaged. Regarding the selected portion, it was the one that allowed us the greatest extractability of phenolic compounds to guarantee greater bioactive potential.